# Early Evidence of the Interplay between Separation Anxiety Symptoms and COVID-19-Related Worries in a Group of Children Diagnosed with Cancer and Their Mothers

**DOI:** 10.3390/children9040481

**Published:** 2022-04-01

**Authors:** Chiara Dotto, Maria Montanaro, Silvia Spaggiari, Valerio Cecinati, Letizia Brescia, Simona Insogna, Livia Zuliani, Paolo Grotto, Cristina Pizzato, Daniela Di Riso

**Affiliations:** 1Department of Developmental and Socialization Psychology (DPSS), University of Padua, 35131 Padova, Italy; chiara.dotto.1@studenti.unipd.it (C.D.); daniela.diriso@unipd.it (D.D.R.); 2Complex Structure of Pediatrics and Pediatric Oncohematology “Nadia Toffa”, Central Hospital Santissima Annunziata, 74121 Taranto, Italy; maria.montanaro@libero.it (M.M.); valerio.cecinati@asl.taranto.it (V.C.); letiziapomponia.brescia@asl.taranto.it (L.B.); simona.insogna@asl.taranto.it (S.I.); 3Simple Structure of Pediatric Oncohematology, Treviso Hospital, ULSS2 Marca Trevigiana, 31100 Treviso, Italy; livia.zuliani@aulss2.veneto.it (L.Z.); paolo.grotto@aulss2.veneto.it (P.G.); cristina.pizzato@aulss2.veneto.it (C.P.); 4Center for Clinical Psychological Services (SCUP), University of Padua, 35121 Padova, Italy

**Keywords:** children with cancer, psychological functioning, COVID-19-related worries, separation anxiety

## Abstract

Having cancer in childhood is a risk factor for separation anxiety symptoms, with negative effects on the disease and psychological wellbeing. The Covid-19-pandemic-related concerns could have a negative effect. The present study explores the interplay between separation anxiety symptoms and COVID-19-related worries in pediatric cancer patients and their mothers, compared to a group of healthy children and their mothers, during the pandemic. Thirty-three subjects with cancer, aged 7–15 years, and their mothers were compared to a control sample of 36 healthy children and caregivers. They were administered a pandemic-related psychological experience survey and standardized questionnaires assessing psychological wellbeing, anxiety, and separation anxiety symptoms. Children with cancer reported significantly higher prosocial behaviors, fear of being alone, and fear of abandonment. Their mothers had worse psychological wellbeing, higher COVID-19 concerns, anxiety, and separation anxiety symptoms. The multiple linear regression model showed that an increase in children’s separation anxiety symptoms was associated with younger age, more recent diagnosis, more mother-child time, lower mothers’ worry for children’s contagion, and higher mothers’ and children’s anxiety. COVID-19-related worries of clinical children’s mothers seem to be protective for children’s psychological wellbeing. Early psychosocial support interventions for mothers are essential in health services for cancer.

## 1. Introduction

Pediatric oncohematological pathologies are a heterogeneous category and the second cause of death between 7 and 15 years old [1]. Between 2016 and 2020, 7000 pediatric oncohematological diseases have been diagnosed in Italy [2]. Having a tumor is a stressful event for a child, with physical and psychological sequelae [3]. Literature reports greater anxiety and separation anxiety in people with tumors [3] or cancer survivors [4] than in healthy children: more frequently, they manifest distress [5] or resistance to separation from their caregivers, especially mothers [4,6]. Children with cancer are at risk for separation anxiety symptoms, due to their life-threatening medical condition, the need for medications, and recurrent hospitalizations [7]. This pattern is further aggravated by COVID-19, which works as a distressing experience [8,9]. Research on pediatric cancer patients during the COVID-19 pandemic revealed the risk of worse outcomes from infection because of immunosuppression [10], less treatment adherence [11], and parents’ increased stress and anxiety [10,12]. Among the main psychological issues related to the COVID-19 pandemic in pediatric patients, particularly concerning are separation anxiety symptoms and COVID-19-related worries [13]. The COVID-19 worries are not supposed to induce separation anxiety symptoms, but to play a crucial role in increasing them [13]. Intense worrying about safety of both children and their close relatives due to the pandemic may lead to difficulties in separation, increased separation anxiety symptoms [13], and a worsening effect on general psychological wellbeing and medical outcomes [13].

Scarce and inconsistent literature is available about the interplay between COVID-19-related worries and separation anxiety in chronically ill children. Italian studies on diabetic children report normative levels of psychological symptoms, but an increase in children’s separation anxiety symptoms during the first lockdown, also associated with higher perceived fear of COVID-19 infection [13]. No differences in psychological functioning emerged when comparing Italian asthmatic children to their healthy peers during the first lockdown, except for higher worry about COVID-19 contagion. Their mothers reported, instead, greater worry about contagion, resumption of activities, and a worsening in psychological wellbeing [14]. Other studies showed worsened children’s psychological wellbeing: pediatric patients with chronic lung diseases and their parents had more pandemic-related concerns and anxiety symptoms [15]. Focusing on mothers’ COVID-19-related worries, the literature showed mixed results, considering them either a protective or a risk factor for the psychological functioning of healthy children. According to some authors, parents most concerned about the pandemic tend to be more protective and to address their children’s emotional responses and worries. Providing containment could make children feel safer, improving their psychological wellbeing [16]. Other studies on healthy samples pointed out that mothers’ COVID-related worries can compromise their ability to recognize and respond adequately to children’s signals, increasing their anxieties and concerns [17]. Very few studies explored the COVID-19-related worries in chronically ill children’s families [14,18]; no studies to our knowledge examined the impact of those parents’ concerns on children’s psychological functioning. 

The present study explored the interplay between separation anxiety symptoms and COVID-19-related worries in a group of Italian children with cancer and their mothers. First, they were compared to a group of healthy children and their mothers; we hypothesized greater fear of COVID-19 contagion [9], psychological difficulties [3], separation anxiety symptoms [4], and anxiety for the clinical group [19]; similarly, their mothers were expected to present greater concerns about COVID-19 [9], psychological distress [20], separation anxiety, and anxiety symptoms [21] than healthy children’s mothers. Moreover, a multiple regression model was performed to evaluate the association between clinical children’s separation anxiety symptoms and COVID-19-related worries, in particular mothers’ and children’s worries about contagion and resumption of activities in a post-lockdown scenario. It was hypothesized that tumor-afflicted children’s separation anxiety symptoms were also associated with their age, time since diagnosis, mother-child time, and mothers’ and children’s anxiety. Separation anxiety is said to be higher in younger children [22]. A diagnosis of cancer may compromise children’s self-regulation of emotions, activating more attachment behaviors, especially in the first period [23]. Time spent with the caregiver is supposed to affect children’s emotional state, as the relationship between mother and child’s psychological functioning is assumed in literature [24].

## 2. Materials and Methods

### 2.1. Subjects

The clinical sample’s children and their mothers were selected from those followed by the Pediatric Oncology Units of Taranto and Treviso Hospitals. The exclusion criteria were comorbidity with psychiatric or other chronic diseases and poor comprehension of Italian. The inclusion criteria were age between 7 and 15 years and at least 2 months from the diagnosis of pediatric cancer. All the families in line with the inclusion criteria who were attending the clinics at the time of the research were presented with the research and they all consented to participate. Thirty-three children (48.5% males, 51.5% females) and their mothers were recruited. Forty-five percent had a hematologic malignancy, 40% a solid tumor, and 15% other hematological pathologies. 

Thirty-six healthy children (44.4% males, 55.6% females), matched by age and gender to the clinical sample, and their mothers, were considered as a control sample. The exclusion criteria were the presence of any pediatric chronic or psychiatric disease, and poor comprehension of Italian. 

Mothers of the two groups did not significantly differ according to age, as the Student’s *t*-test demonstrated (*p* > 0.05). Most of the clinical sample’s mothers were housewives, while mothers of the control sample were mainly part-time or full-time workers at the time of the compilation, as shown in Table 1. 

### 2.2. Procedure

Pediatric cancer patients’ data were collected between November 2020 and May 2021. Participation was proposed at least 2 months after diagnosis, considering the acute stress that occurs immediately after diagnosis [23]. The thirty-three families in line with the inclusion criteria who were attending the Pediatric Oncology of Taranto and Treviso at the time of the research were contacted to participate. Mothers were introduced to the research, its objectives, and modality during the outpatient control visit, and asked to sign a detailed informant consent form to join the survey, authorizing themselves and their children to participate. A specific informant consent form was prepared for 12- to 15-year-old children, according to the Ethics Committee’s requirements. In any case, all children were verbally informed about the study and asked if they wanted to join the survey. All 33 mothers gave their consent for their study participation. As for the 33 children, informed consent was obtained from a parent or legal guardian, and from themselves if 12-to-15 years old. Children with cancer and their mothers filled out the questionnaire online (via an e-mail link).

To begin with, a survey created by the researchers was proposed: mothers were asked about their sociodemographic characteristics (e.g., age, current working situation), their worries about the COVID-19 pandemic (e.g., concerns about their or their children’s contagion, the resumption of children’s activities after the COVID-19 emergency), and time, in hours, spent on average with their child in a working day. Some cancer issues, such as the type and time since diagnosis, were included. Children were asked about their worries related to the COVID-19 pandemic.

Lastly, both were required to complete standardized self-report questionnaires, which assessed psychological adjustment (Strengths and Difficulties Questionnaire, SDQ), separation anxiety (Separation Anxiety Assessment Scale for Children, SAAS-C), and anxiety (State-Trait Anxiety Scales Inventory for Children, STAI-C) in children and general wellbeing (General Health Questionnaire, GHQ-12), separation anxiety (Adult Separation Anxiety Questionnaire, ASA-27), and anxiety (State-Trait Anxiety Inventory, STAI) in mothers, respectively. The control sample was recruited through snowball sampling; the CESC guidelines for administration were also followed for the recruitment of the control sample. The procedure and the data collection period were the same as for the clinical sample; parents were contacted and were met with personally to be informed about the research. COVID-19 safety guidelines were followed. The surveys were almost identical, except for the items related to oncohematological pathologies. The child’s compilation took about 20 min, while the parents’ one took about 30/40 min. The project was approved by the Ethics Committee for Clinical Trials (CESC) (Observational study n. 977/CE). CESC approval’s guidelines are in line with the Ethical and Deontological Codes of Italian Psychologists. No reward was offered for enrollment.

### 2.3. Measures

Children’s psychological functioning. The Strengths and Difficulties Questionnaire (SDQ) [25] is a 25-item screening tool for children’s behavioral and emotional problems. The 3-point Likert scale ranges from 0 (“not true”) to 2 (“certainly true”). It consists of 5 subscales: emotional symptoms, conduct problems, hyperactivity and inattention, peer problems, and prosocial behaviors. A total difficulties score is obtained by adding the first four scales, which measure adjustment difficulties. The prosocial behavior scale assesses whether children are kind and whether they share and help others. Moreover, internalizing and externalizing symptom scales can be used. The total score values for the clinical range are above 20. The Italian version demonstrated good validity and reliability and has been validated for 8-to-15-year-old children and adolescents [26]. Cronbach’s alpha values for the total score (TDS), the internalizing symptoms scale (INT), and the externalizing symptoms scale (EXT) were α(TDS) = 0.78, α(INT) = 0.60, α(EXT) = 0.71.

Separation Anxiety Assessment Scale for Children (SAAS-C) [27] is a 34-item self-report questionnaire to assess children’s separation anxiety symptoms. Each item is evaluated on a 4-point Likert scale from 1 (“never”) to 4 (“always”). The six scales are fear of abandonment, fear of being alone, fear of physical injuries, worry about calamitous events, frequency of calamitous events, and safety signal index. The latter two are not considered diagnostic of separation anxiety disorder, thus they were not used in the present research [28]. The Italian version’s validity and reliability have been proven [29]. Cronbach’s alpha value for the considered four scales was α(SAAS-TOT) = 0.85.

State-Trait Anxiety Inventory for Children (STAI-C) [30] is a self-report questionnaire providing a measure of children’s anxiety. It consists of two independent scales of 20 items each: the state scale assesses children’s current feelings of anxiety, asking how they are feeling; the trait scale assesses the tendency to experience anxiety, asking how they generally feel. All items are rated on a 3-point Likert scale. Scores above 32 and 40 indicate clinical range, respectively, for the state and trait scales. The Italian version’s validity and reliability have been proven [31]. Cronbach’s alpha value for the total score was α(STAI-C-TOT) = 0.92, and for the state and trait scale it was α(STAI-C-S) = 0.92 and α(STAI-C-T) = 0.86, respectively.

Mothers’ psychological functioning. The General Health Questionnaire (GHQ-12) [32] is a screening tool to assess adults’ short-term psychological wellbeing. It consists of 12 items rated on a 4-point Likert scale, from 0 (“more than usual”) to 3 (“much less than usual”). Three score ranges can be distinguished: normal (lower scores), psychological suffering, and important difficulties (scores above 20), which could require professional intervention. The validity and reliability of the Italian version have been confirmed [33]. Cronbach’s alpha was α(GHQ-TOT) = 0.70.

The Adult Separation Anxiety Questionnaire (ASA-27) [34] is used to assess separation anxiety in adults. It has 27 items rated on a 4-point Likert scale, from 0 (“never”) to 3 (“very often”). A total score is obtained by adding all items’ scores: values for the clinical range are above 22. The Italian version’s validity and reliability have been confirmed [35]. Cronbach’s alpha was α(ASA-TOT) = 0.90.

The State-Trait Anxiety Inventory (STAI) [36] is a self-report scale assessing state and trait anxiety in adults. As in the children’s version, 20 items evaluate trait anxiety and 20 evaluate state anxiety. Raters respond on a 4-point Likert scale. Scores above 41 and 53 indicate clinical range, respectively, for the state and trait scales. The Italian version has good validity and reliability [37]. Cronbach’s alpha value for the total score was α(STAI-M-TOT) = 0.97, and for the state and trait scale was α(STAI-M-S) = 0.96 and α(STAI-M-T) = 0.93, respectively.

### 2.4. Data Analysis

Shapiro–Wilk test was performed for testing the normality of the sociodemographic and psychological variables: all of them are non-normally distributed. To evaluate the differences between the clinical and control groups in psychological variables and standardized questionnaire scores (SDQ, SCAS, STAI-C, GHQ, ASA-27, and STAI), the Mann–Whitney U test was used. Multiple linear regression analysis with pediatric cancer patients’ separation anxiety symptoms as the dependent variable was performed. The stepwise method was used. For all the analyses, a *p*-value < 0.05 was considered statistically significant. The SPSS v22.0 software package (SPSS Inc., Chicago, IL, USA) was used for the statistical analysis.

## 3. Results

### 3.1. Differences between Clinical and Control Children in Selected Variables of the Ad Hoc Survey and Standardized Questionnaires

Ninety-three percent of the children with cancer reported non-clinical scores in the SDQ. For some selected variables of the online survey, such as children’s concern about COVID-19 contagion and the resumption of their activities, no statistically significant differences emerged between the clinical and control groups (*p* > 0.05). According to standardized questionnaires, pediatric cancer patients reported higher scores in the SDQ “prosocial behavior scale” (*p* < 0.05), and in the SAAS-C “fear of being alone” and “fear of abandonment”, than healthy control with medium effect sizes (*p* < 0.05). Lastly, no significant differences were found in STAI-C between the two groups (*p* > 0.05) (Table 2).

### 3.2. Differences between Clinical and Control Mothers in Selected Variables of the Ad Hoc Survey and Standardized Questionnaires

Fifty-one percent of the clinical group’s mothers reported psychological suffering and 33% showed the need for possible intervention, according to GHQ scoring. As regards some selected variables of the online survey, mothers of the clinical sample reported more concerns about their children’s contagion and the resumption of their children’s activities. Moreover, mothers of pediatric cancer patients spent more time with their children in a working day (*p* < 0.05). According to standardized questionnaires, mothers of the clinical group reported higher scores in the GHQ, the ASA-27, the STAI total score, and both trait and state anxiety compared to healthy control mothers, with medium to large effect sizes (*p* < 0.05) (Table 3).

### 3.3. Predictors of Cancer-Afflicted Children’s Separation Anxiety Symptoms

The multiple linear regression model showed that an increase in cancer-afflicted children’s separation anxiety symptoms was associated with younger age, more recent diagnosis, more mother-child time in a working day, lower mothers’ worry about children’s contagion, higher mothers’ total anxiety, and children’s trait anxiety (Table 4). Regression model diagnostics indicated no problems with multicollinearity; the data met the assumption of independent errors (Durbin–Watson value = 1.895), and the Shapiro–Wilk test of standardized residuals indicated that the data contained normally distributed errors (*p* = 0.367). Moreover, it was verified that the assumptions of linearity and homoscedasticity were met by analyzing the plots. 

## 4. Discussion

The present study explored the interplay between separation anxiety symptoms and COVID-19-related worries in a group of Italian children with cancer and their mothers. As to the children, patients with cancer did not significantly differ from healthy peers in COVID-19-related worries, general psychological functioning, and anxiety symptoms, in line with the literature that reports unchanged psychological functioning for children with chronic diseases such as diabetes and asthma in the first lockdown [13,14]. A study on Dutch children with cancer observed no differences in health-related quality of life and fatigue between the pre- and early-COVID-19 periods; authors referred to children’s experience in dealing with medical traumatic events and adequate care and support received to explain these results [38]. The present sample consisted of children who were not at risk for psychopathological diseases and the medical clinics involved pay close attention to the psychological aspects, involving psycho-oncologists, giving reassuring information and continuous medical monitoring. However, consistent with the hypothesis and the literature [3,13], children with cancer showed higher separation anxiety symptoms. Separation anxiety in pediatric cancer patients is a typical feature. The life-threatening condition, the need for medical monitoring, and parents’ anxiety may make children more afraid of separation [7]. Lastly, pediatric patients reported more prosocial behaviors than healthy children, which is a potential resiliency factor. Caring for others can reduce social isolation, which is prevalent in such a population because of health problems and is exacerbated by the pandemic [21]. In addition, the changes imposed by COVID-19, such as homeschooling and increased hygiene awareness, may decrease their feelings of being different from others [38]. These results should be further explored in future research because literature about children with cancer mainly focuses on psychological weaknesses instead of resources.

Regarding mothers, the clinical group showed greater COVID-19-related worries, worse psychological wellbeing, higher anxiety, and separation anxiety symptoms, compared to healthy children’s mothers. They also reported more mother-child time in a working day. These results are in line with the hypothesis and the literature available to date: chronically ill children’s parents indicated greater concerns about contagion and worse psychological wellbeing when compared to healthy children’s parents [14,15]; pediatric cancer patients’ mothers reported clinical levels of anxiety during the pandemic [12]. Mothers must provide continuous assistance and might experience traumatic events completely on their own, also because the hospital allows only one parent to be present during the outpatients’ visits and the hospitalizations as prevention measures. They need to screen out emotional triggers for their children, improving their psychological wellbeing, but they are struggling hard, eventually suffering from a dual trauma, due to the diagnosis of cancer and the sense of vulnerability related to COVID-19. High levels of physical and psychological stress [39] and a worsening of preexisting situations of imbalance can occur. 

Considering the greater rates of separation anxiety found in children with cancer, possible associated factors were explored. Separation anxiety may be considered a multifaceted variable, probably associated with factors related to the disease and psychological functioning, such as mothers’ and children’s COVID-19-related worries and anxiety. The present study showed that higher levels of children’s separation anxiety symptoms were associated with younger age, more recent diagnosis, and more mother-child time in a working day, in line with the hypothesis and the literature [22,23]. As said before, the risky condition of having cancer may be linked to children’s separation anxiety [7]. Moreover, anxious mothers may feel the need to spend more time with their children, perceiving less control on time management, increasing stress and the child’s separation anxiety [21,24]. As to the psychological functioning of children with cancer and their mothers, curiously, higher separation anxiety symptoms were associated with lower mothers’ worry about children’s contagion. It is not clear in the literature if COVID-19 concerns are a protective or risk factor for the psychological wellbeing of mothers and children. The present results support the thesis that COVID-19-related worries are protective for children’s psychological functioning. Mothers more concerned about contagion may adopt behavioral and emotional containment strategies, thereby reducing children’s separation anxiety symptoms [16]. Pediatric patients’ COVID-19-related worries seem not to directly influence separation anxiety symptoms, contrary to the hypothesis. Clinical children in the present study showed normative psychological functioning and similar levels of COVID-19-related worries and anxiety as healthy peers. Moreover, the maternal role is crucial in these psychological dynamics. Lastly, higher children’s separation anxiety symptoms were associated with greater mothers’ and children’s anxiety, in line with the considerable body of literature assuming an association between parents’ mental health and children’s psychological development [21,23,24].

The present research has some limitations: the small sample size, also due to the short timeframe for collecting data, the unavailability of a baseline of children’s and mothers’ psychological functioning before the pandemic, the lack of fathers’ involvement, and the difficulty of homogeneous division of patients by diagnosis must be considered. Furthermore, no information about how care was divided between parents or how the child spent time during the day while mothers were working was collected. Moreover, the use of self-report tools may have affected the results. Nonetheless, this study explores a very new topic and contributes to highlighting the association between COVID-19-related worries and chronically ill children’s separation anxiety symptoms.

The present paper showed that pediatric cancer patients’ mothers have worse psychological outcomes as to the pandemic, negatively influencing their children’s separation anxiety symptoms. New challenges for the health services are emerging, such as the need to provide multidisciplinary intervention both for the parents, especially mothers, and the child, and continued monitoring of their psychological state. Future studies may involve other family members, consider different types of diagnosis and different stages of the disease, and monitor the psychological sequelae of the pandemic over time.

## 5. Conclusions

Our findings support the hypothesis that mothers’ COVID-19-related worries are protective for their children’s psychological wellbeing, as associated with separation anxiety symptoms. Mothers’ anxiety may cause them to pay more attention to their children, trying to protect them and keep them from worrying. However, that drives mothers to weaken. Thus, early intervention for mothers’ psychological support may contribute to improving care for this population. 

## Figures and Tables

**Table 1 children-09-00481-t001:** Demographics and descriptive variables for the clinical and control group (means, standard deviations, and percentages). Student’s *t*-tests to compare the two groups are reported.

		Clinical Sample (N = 33)		Control Sample (N = 36)		*t*-Test/χ^2^ Test	*p*-Value
Children		Mean/Percentage	SD	Mean/Percentage	SD		
Age		10.78	3.15	10.86	2.48	602.500	0.743
Gender	Males	48.5%		44.4%			
	Females	51.5%		55.6%			
Mothers		Mean/Percentages	SD	Mean/Percentages	SD	*t*-test/χ^2^ test	*p*-value
Age		41.76	6.205	43.75	4.66	1516	0.134
Working Situation	Housewife	37.5%		22.2%		11,041	0.050
	Unemployed	15.6%		5.6%			
	Work part-time outside	21.9%		41.7%			
	Work full-time outside	21.9%		27.8%			
	Smart working part-time	3.1%		2.8%			

**Table 2 children-09-00481-t002:** Mann–Whitney U tests used to evaluate the differences between children of the clinical and control groups in survey variables and standardized questionnaires. Mean rank, minimum (min) and maximum (max), standard deviations, and test statistics are reported. Effect sizes’ r values: 0.00–0.20 low, 0.20–0.50 medium, 0.50–2.00 large.

	Clinical Sample (N = 33)		Control Sample (N = 36)		Test Statistic	*p*	Effect Size r
	Mean Rank (Min-Max)	SD	Mean Rank (Min-Max)	SD			
Children’s concerns about contagion	34.68 (1–3)	0.79	33.42 (1–3)	0.61	579.000	0.772	0.04
Children’s worry about the resumption of their activities	30.87 (1–3)	0.67	36.69 (1–3)	0.71	461.000	0.182	0.16
SDQ Total score	32.88 (2–29)	5.90	34.01 (1–26)	4.80	521.500	0.811	0.02
SDQ Internalizing symptoms	34.47 (0–12)	2.98	32.69 (0–8)	2.14	569.000	0.706	0.10
SDQ Externalizing	31.90 (0–13)	3.02	35.81 (0–15)	2.79	493.000	0.410	0.08
SDQ Emotional symptoms	33.03 (0–10)	2.43	33.89 (0–6)	1.89	526.000	0.855	0.02
SDQ Conduct problems	34.31 (0–6)	1.36	33.74 (0–7)	1.50	567.500	0.902	0.01
SDQ Hyperactivity/inattention	31.27 (0–9)	2.37	36.35 (0–10)	2.02	473.500	0.283	0.12
SDQ Peer problems	34.40 (0–7)	1.72	33.65 (0–5)	1.46	570.500	0.871	0.04
SDQ Prosocial behaviors	39.06 (2–10)	1.82	29.64 (4–10)	1.74	715.000	0.045	0.20
SAAS-C Total score	36.69 (41–91)	14.27	31.68 (39–91)	1.58	641.500	0.293	0.08
SAAS-C Fear of Being Alone	41.40 (5–17)	2.89	27.63 (5–17)	3.10	787.500	0.003	0.26
SAAS-C Fear of Abandonment	39.31 (5–13)	2.04	29.43 (5–9)	1.32	722.500	0.024	0.24
SAAS-C Fear of Physical Injuries	38.37 (5–14)	2.81	30.24 (5–11)	1.76	693.500	0.080	0.26
SAAS-C Worry about Calamitous Events	30.23 (5–18)	3.68	37.25 (6–18)	3.50	441.000	0.138	0.14
STAI-C Total score	33.78 (45–101)	15.74	32.38 (44–83)	8.91	499.500	0.766	0.05
STAI-C Trait anxiety	31.74 (21–54)	7924	34.01 (23–53)	6.29	485.500	0.630	0.01
STAI-C State anxiety	38.18 (23–54)	8.864	30.40 (21–41)	4.77	687.500	0.103	0.27

**Table 3 children-09-00481-t003:** Mann–Whitney U tests used to evaluate the differences between mothers of the clinical and control groups in survey variables and standardized questionnaires. Mean rank, minimum (min) and maximum (max), standard deviations, and test statistics are reported. Effect sizes’ r values: 0.00–0.20 low, 0.20–0.50 medium, 0.50–2.00 large.

	Clinical Sample (N = 33)		Control Sample (N = 36)		Test Statistic	*p*	Effect Size r
	Mean Rank (Min-Max)	SD	Mean Rank (Min-Max)	SD			
Mothers’ worry about children’s contagion	43.66 (1–3)	0.61	26.36 (1–3)	0.66	869.00	0.012	0.47
Mothers’ worry about the resumption of children’s activities	42.30 (1–3)	0.69	27.57 (1–2)	0.51	825.50	0.001	0.42
Mothers’ time with the child in a working day	42.02 (2–24)	6.09	23.90 (0–10)	3.27	741.50	0.000	0.52
GHQ-12 Total score	42.94 (3–28)	4.88	27.72 (6–26)	4.81	856.00	0.002	0.33
ASA-27 Total score	39.81 (7–56)	13.42	29.00 (9–41)	8.53	738.00	0.023	0.31
STAI Total score	44.63 (46–149)	23.78	25.50 (51–122)	16.12	900.00	0.000	0.46
STAI Trait anxiety	43.44 (23–75)	11.26	27.26 (25–67)	8.74	872.50	0.001	0.38
STAI State anxiety	44.30 (23–77)	14.03	25.79 (24–55)	8.59	889.50	0.000	0.47

**Table 4 children-09-00481-t004:** Multiple linear regression model of cancer-afflicted children’s separation anxiety symptoms (*n* = 33). B, unstandardized beta; Std. β, standardized beta; CI, confidence intervals; Adj. R2, adjusted R-squared; *p* < 0.05 in bold.

	Children’s Separation Anxiety Symptoms			
	B (95% CI)	Std. β	t	*p*
Intercept	63.255 (24.345, 102.165)		3.542	0.004
Children’s age	−2.742 (−4.238, −1.246)	−0.528	−3.993	0.002
Time since diagnosis	−0.613 (−1.011, −0.214)	−0.512	−3.348	0.006
Mothers’ time with the child in a working day	1.290 (0.179, 2.402)	0.396	2.529	0.026
Mothers’ worry about children’s contagion	−13.112 (−25.104, −1.119)	−0.465	−2.382	0.035
Mothers’ worry about the resumption of children’s activities	−5.234 (−14.524, 4.056)	−0.197	−1.227	0.243
Children’s concerns about contagion	−0.408 (−6.928, 6.113)	−0.020	−0.136	0.894
Children’s worry about the resumption of their activities	2.609 (−4.256, 9.474)	0.114	0.828	0.424
STAI Total score	0.221 (−0.001, 0.444)	0.313	2.166	0.050
STAI-C Trait anxiety	1.471 (0.800, 2.143)	0.668	4.772	0.000
Model fit	F(9,12) = 7.197			
	*p* = 0.001			
Adj. R^2^	0.726			

## Data Availability

The datasets generated and used for the present study are available from the corresponding author upon reasonable request.

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
