# Peer review of "Early Evidence of the Interplay between Separation Anxiety Symptoms and COVID-19-Related Worries in a Group of Children Diagnosed with Cancer and Their Mothers"

_children, 2022, doi:10.3390/children9040481_

Round 1

Reviewer 1 Report 

The article is interesting and the topic is highly relevant, especially for this specific population category. However, there are some aspects that should be reconsidered in a revision: 

  1. Language and syntax (e.g., "Clinical sample’s children and their mothers were selected between those followed 91 by the “Nadia Toffa” Pediatric Oncology Unit of the Hospital “SS. Annunziata” in Taranto 92 and by the Pediatric Oncohematology of the ULSS 2 - Marca Trevigiana in Treviso" - this a confusing statement that should be revised).
  2. Why did you choose this specific age group? i.e., age 7 to 15. Why not younger, or why not older?
  3. This is an incorrect statement: Mothers of the two groups did not differ according to age, as the Student t-test 105 demonstrated (p>.05). They did differ, but not statistically significant. Please revise.
  4. Why did the other children did not receive "A specific informant consent", and it was only prepared for 12- to 15-year-old children? Is there a specific reason for this? Did you even ask the children if they wanted to participate? I am not sure the research follows the necessary ethical guidelines. There also no mentions considering the ethical requirements. 
  5. Since the dependent variable/criterion was not normally distributed, how and why did you choose regression analysis? Did you check for the necessary statistical conditions for the regression? If not, the entire results section needs to be rewritten, since normality is a necessary condition for regression analysis. 
  6. The Limitations section needs to be improved. There is too little information considering the study's various limitations, especially the very small sample size.

Reviewer 2 Report

Dear authors,

The paper on the interplay between separation anxiety and Covid-19 related worries in in children and their mother is an interesting paper, which addresses a complex issue. The paper is well written and easily readable. However, in my opinion, there are some issues that need to be addressed before publication.

My main concerns are on the added value of Covid-19 related worries. The rationale on how Covid-19 related worries could induce separation anxiety remains unclear throughout the manuscript. In lines 46-48, the authors state that these are particularly appealing, but there is no reference to prior literature supporting this statement. Additionally, it is not quite clear why this should be investigated in children with cancer specifically. Studying separation anxiety in this group seems very logical to me, but not how Covid-19 related worries could be such an important variable in this association.

The concerns about the relevance of the Covid-19 worries remained throughout the whole manuscript. In the methods section it appears that only a few questions were included in the questionnaires on worries related to the pandemic. Did the authors consider to include questions on the comparison of specific variable before and during the pandemic? For example; did you experience these feelings in the same way before Covid-19 pandemic?

In the results/discussion section, the authors state (lines 255-259) that the separation anxiety might be a result of the cancer rather than the pandemic, but this is not further elaborated on, while to me, this seems to be exactly the point that should be addressed. I think it is what you’d expect; separation anxiety is higher in children with cancer, and is probably associated with age, time since diagnosis, etc. The emphasis however, is on Covid-19 related worries, and this doesn’t seem quite right to me.

The conclusion is completely focused on this, and even recommends early interventions on psychological support for mothers on their pandemic-related worries. The evidence for how these worries might be protective for the child’s psychological well-being is however quite thin. Also, it might be highly correlated with the association that children with experience higher anxiety separation when their mothers spend more time with them during the day. It might be hypothesized that in these families, life as usual goes on, which leaves the child feel insecure on how important this diagnosis is to his/her family. Additionally, if lower maternal worries induce higher anxiety in the child, why would you recommend psychological support to the mothers? Wouldn’t that further decrease the maternal worries and therefore increase separation anxiety in the child?

Some other remarks;

  • In the introduction, the part on separation anxiety in children with cancer should be discussed earlier.
  • Lines 69-71; when the authors state that there are a few studies, I’d expect more than one reference. The same goes for ‘even less studies’ and no references.
  • In 2.1, the authors describe that specific groups were not included. This sounds like they were excluded after agreeing to participate.
  • 2.1/2.2, how were healthy children recruited?
  • 2.2, in line 119 the authors describe an informant consent for children 12-15, in lines 121-122 they speak about children older than 12.
  • 2.1/2.2, please add information on the power.
  • 2.3, SDQ consists of five subscales, but the 5th was not included in the total score. Please explain. Also, prosocial behaviour should be explained a bit further in the text.
  • 2.3, the authors explain that two scales were excluded from the SAAS-C. Did you consider the validity issues when excluding two scales?
  • 2.3, when using a total score, it might be helpful and informative to add the cut-off points for high and low scores. For the STAI for example, the mean scores of mothers in the clinical sample are slightly above cut-off, while they are below cut-off for the control sample (table3).
  • For all tables; please add min-max of all questionnaires in the explanatory text above the table.
  • Discussion, lines 299-300; only one reference after stating that there is a considerable body of literature seems a bit thin.
  • Limitations; were questions asked about how care was divided between fathers and mothers? Or where the child was during the day when mothers were working?
  • Lines 306-307; this statement is a bit too strong. The authors did not study causality on psychological outcomes directly caused by the pandemic. Please rephrase.
